# Formulation Strategies for Enhancing the Bioavailability of Silymarin: The State of the Art

**DOI:** 10.3390/molecules24112155

**Published:** 2019-06-07

**Authors:** Alfonso Di Costanzo, Ruggero Angelico

**Affiliations:** 1Centre for Research and Training in Medicine for Aging, Department of Medicine and Health Sciences “Vincenzo Tiberio”, University of Molise, I-86100 Campobasso, Italy; alfonso.dicostanzo@unimol.it; 2Department of Agriculture, Environmental and Food Sciences (DIAAA), University of Molise, I-86100 Campobasso, Italy

**Keywords:** silymarin, silybin, nanoemulsion, solid lipid nanoparticles, nanostructured lipid carriers, liposome, polymeric particles, self-emulsifying delivery systems, enhanced bioavailability

## Abstract

Silymarin, a mixture of flavonolignan and flavonoid polyphenolic compounds extractable from milk thistle (*Silybum marianum*) seeds, has anti-oxidant, anti-inflammatory, anti-cancer and anti-viral activities potentially useful in the treatment of several liver disorders, such as chronic liver diseases, cirrhosis and hepatocellular carcinoma. Equally promising are the effects of silymarin in protecting the brain from the inflammatory and oxidative stress effects by which metabolic syndrome contributes to neurodegenerative diseases. However, although clinical trials have proved that silymarin is safe at high doses (>1500 mg/day) in humans, it suffers limiting factors such as low solubility in water (<50 μg/mL), low bioavailability and poor intestinal absorption. To improve its bioavailability and provide a prolonged silymarin release at the site of absorption, the use of nanotechnological strategies appears to be a promising method to potentiate the therapeutic action and promote sustained release of the active herbal extract. The purpose of this study is to review the different nanostructured systems available in literature as delivery strategies to improve the absorption and bioavailability of silymarin.

## 1. Introduction

The number of scientific reports in the literature testifying to the efforts of experts in the fields of colloidal and pharmaceutical sciences in proposing silymarin-based formulations with enhanced stability and solubility, improved bioavailability and efficient therapeutic performances is constantly growing. Silymarin, an official medicine with recognized hepatoprotective, antioxidant, antiviral and anticancer properties, suffers from poor water solubility, inefficient intestinal absorption, presence of multiple components and elevated metabolism [1,2]. In general, these undesired properties pose several concerns in the formulation and design of drug delivery systems since inadequate aqueous solubility of the active ingredients constituting the phytomedicines can affect the correspondent bioavailability [3]. On the other hands, a wide range of solubilization technologies is now available for researchers to entrap poorly water-soluble active plant ingredients such as silymarin in aqueous nanovehicles [4].

For instance, an efficient nanoencapsulation allows phenolics and antioxidants to be absorbed passively from the lumen of the intestine into the lymphatic and blood circulatory system; therefore, their bioavailability can be notably increased [5,6]. Therefore, silymarin-based formulations have begun to be prepared with innovative techniques and have been found to increase its therapeutic efficacy against various diseases [7].

Several reviews have appeared on the subject in the recent past, each of them trying to offer the reader a broad and increasingly updated vision of the most advanced solutions developed to increase the in vivo absorption efficiency of silymarin [8,9,10,11,12]. These include drug modifications such as salts, esters and complexes with hydrophilic excipients have been used to optimize silymarin solubility. Most of the formulation strategies refer to complexation with cyclodextrins or phospholipids (phytosomes), solid dispersions stabilized by biocompatible polymers, micro- and nanoemulsions, lipid-based delivery systems, biodegradable polymeric NPs and inorganic nanomaterials.

Self-emulsifying drug delivery systems (SEDDS) have attracted interest in the pharmaceutical industry because they can be dosed as preconcentrates and generate a drug-containing emulsion with a large surface area upon dispersion in the GIT. When used in SEDDS, surfactant molecules in a preconcentrate formulation self-assemble to form nanoparticles containing the drug. Thus, to favour faster absorption, higher concentrations of biocompatible amphiphilic compounds are used to form smaller particles. Another technological approach consists in generating a supersaturated solution, characterized by increased chemical potential compared to a stable, saturated solution. Supersaturation, therefore, is a thermodynamically unstable condition and the system tends to return to the equilibrium state by drug precipitation.

Thus, to apply the concept of supersaturation in order to circumvent the problem of the limited water solubility of silymarin and increase its absorption in the stomach and small intestine, the supersaturated solution must be stabilized through the use of precipitation inhibitors for a sufficient period of time. For example, different degrees of supersaturation and maintenance of the supersaturated state can be achieved using amorphous solid dispersions with different polymers such as PVP and PVA. On the other hands, controlled-release systems would represent one option to avoid excessive supersaturation. Therefore, advanced versions of these formulation strategies provide the ability to control the release profile of the drug in response to external stimuli such pH, ionic strength, magnetic field, ultrasounds, light and heat, which can improve the targeting and allow greater dosage control.

It is also worth noting that the physicochemical properties of the nanocarriers, such as size, shape, surface chemistry, porosity and elasticity and their interactions with biological barriers determine the therapeutic success of the active ingredients loaded therein [13,14,15]. Therefore, the correct selection of formulation technique and compositional materials to design a suitable delivery system for silymarin play an important role in developing a functional, safe and marketable product eventually. The ambition of the present paper is to summarize and highlight the advances made over the past 15 years regarding the formulation strategies to improve the bioavailability of silymarin, as well as focusing on the emerging trends of the nanodelivery of this drug in new pharmaceutical applications.

## 2. Silymarin: Source and Physicochemical Properties

Silymarin consists of several flavonoid-like compounds extracted from the small hard fruits (kenguil seeds) of *Silybum marianum* L. Gaertn (milk thistle), which grows extensively in Europe and Asia, including India. The drug belongs to a class of compounds—flavonolignans—likely produced in the plant by radical coupling of flavonoids and coniferyl alcohol [16]. A reference composition comprises: silybin or silibinin (33.4%), silychristin (12.9%), silydianin (3.5%) and isosilybin (8.35%) [17,18], which are assumed to be responsible for the therapeutic liver-protecting activity of the plant extract [19,20,21,22]. The silymarin composition and the ratios of individual constituents may vary with respect to different brands of commercial standardized milk thistle extracts [23].

For the main component silybin, a large amount of pharmaco-toxicological and clinical documentation exists [24,25,26]. The molecular structure representative of one of the two diastereoisomers (silybin A and silybin B, 1: 1 mixture) is shown in Figure 1.

Its molecular structure possesses a chromone fragment responsible for its weak acidic properties, enabling donor-acceptor interactions with bases. The presence of polyphenol hydroxyls and the ability to form complexes with transition and other metal ions in the 3,4- or 4,5-positions, confer high antioxidant capability to the molecule. Several studies have shown that for this active ingredient even very high doses are well tolerated by animals and humans [27,28].

In particular, the oral 50% lethal dose is 10,000 mg/kg in rats while the maximum tolerated dose is 300 mg/kg in dogs [29]. Due to its lipophilic nature characterized by a log*P* value of 1.41 [30] where *P* is the partitioning coefficient of the drug, the therapeutic efficiency of silybin is rather limited by its very low water solubility (430 mg/L) [31,32,33,34]. A recent study reported the solubility of silymarin in water, ethanol, isopropyl alcohol, β-CD and PEG-400, the latter having recorded the highest mole fraction solubility at 298 K (0.243) [35]. Another limiting factor consists in a low intestinal permeability, which combined with poor water solubility, categorizes silymarin as a class IV compound, according to the Biopharmaceutical Classification System (BCS) [36].

Consequently, the drug is poorly absorbed (20–50%) from the GIT and has a low bioavailability from oral formulations [37]. Several semisynthetic compounds have been designed to overcome the drawback of very low water solubility such as, e.g., the bis-hemisuccinate (Legalon^®^), 23-*O*-phosphate, 23-*O*-β-glycosides derivatives, silybinic acid and the complex with the excipient amino-sugar meglumine [38,39]. However, chemical modifications leading to an increase in silybin water-solubility usually led to an impairment of its antioxidant (antiradical) activity [31].

The hepatoprotective mechanism of silymarin and its main component silybin is due not only to the antioxidant activity but also to a membrane-stabilizing action that prevents or inhibits the lipid peroxidation process [40].

## 3. Formulation Strategies Designed to Improve the Bioavailability of Silymarin

In the following subsections, various approach specifically designed for silymarin delivery will be reviewed and systematically classified in the final Table 1, including the suitable cross references mentioned in the text. Figure 2 illustrates the types of formulations that have been developed to improve the solubility and bioavailability of silymarin, most of which being designed primarily for oral administration. Henceforth, the acronyms SLM and SIL will be used, respectively, to indicate the silymarin powdered extract and its main purified active constituent silybin or silibinin. The meaning of the other acronyms is shown in the list of abbreviations at the end of the review.

### 3.1. Nanocrystals, Nanosuspensions and Solid Dispersions

Drug nanosuspensions are sub-micron colloidal dispersions of pure drug particles, which are stabilized by surfactants or polymeric steric stabilizers [41]. Nano- and micronization technologies improve the oral bioavailability and dissolution rates and prolong the half-life of sparingly soluble drugs. For instance, Zhang et al. [42] applied the ESD method to produce uniform SIL nanospheres with a mean size of ~240 nm as well as micronized rod-shaped and spherical particles obtained by controlling the temperature and SDS concentration.

X-ray powder diffraction (XRPD) investigations demonstrated a low crystalline state for the rod-shaped smaller particles, which in turn manifested a better dissolution property than the larger spherical ones. However, although the precipitation technology is simple and cost effective, the tendency of the pharmaceutical particles to grow, and the difficulty in inhibiting that growth, posed obstacles to their production at industrial level.

Another application of the nanoemulsification technique with PVP as carrier to produce highly bioavailable SIL NPs with mean size < 200 nm and EE > 97% was reported by Liu et al. [43], to test in vitro the antiviral activity against HCV infection in human hepatocytes. Besides, pharmacokinetic studies in rodents revealed an enhanced serum level and superior drug biodistribution to the liver after oral administration of SIL-loaded NPs compared with bare SIL.

Wang et al. [44] used the HPH technology to design SIL nanosuspension formulations for oral and i.v. administrations with different particle sizes. XRPD and differential scanning calorimetry (DSC) experiments showed that the crystalline structure of SIL was not perturbed as a result of the homogenization and freeze-drying processes. In vitro dissolution profiles and solubility tests of various nanosuspensions, including unmilled commercial SIL and a physical mixture for comparison, yielded an increase in solubility and dissolution rate following the reduction of particle size as predicted by the Noyes-Whitney equation [45]. Such nanosuspensions were able to improve the permeability of the transport of SIL across the Caco-2 cell monolayer. In vitro results were further confirmed by the pharmacodynamics and tissue distribution studies in beagle dogs and mice [46].

An increase of SLM aqueous solubility by almost 650-fold compared to bare drug powder was achieved by incorporating the drug into a solid dispersion, prepared by spray-drying SLM aqueous suspensions in presence of a surfactant-polymer mixture [47].

This strategy, besides using water instead of the organic solvent, led to a reduced ratio of hydrophilic polymeric carrier and drug in the final solid dispersion. A series of 1% aqueous solutions of biocompatible polymers and surfactants were tested to evaluate their capacity as SLM carriers in enhancing the aqueous solubility of the drug. PVP and Tween 80 provided the highest drug solubility of about 800 mg/mL and 2500 mg/mL, respectively, and were selected for preparing SLM-loaded solid dispersions. The optimised SLM/PVP/Tween 80 formulation (5:2.5:2.5, w/w/w), combined both relatively smaller amounts of carriers and increased drug solubility and dissolution. Compared to a commercial product, the proposed solid dispersion improved the oral bioavailability of the drug in rats by almost 3-fold and also exhibited advanced hepatoprotective bioactivity.

SLM/PVP solid dispersion prepared by a fluid-bed coating technique, was also designed as payload of a monolithic osmotic tablet consisting of a tablet core and semi-permeable coating of cellulose acetate with mechanically perforated release orifices [48,49]. This delivery system permitted the synchronized release of the multiple components of SLM in solubilized state under the driving force of the osmotic pressure. In vitro release profiles indicated that the overall release rate was adjustable by monitoring the formulation variables.

In a recent application of the spray-drying technique, NaCMC has been selected as encapsulant agent in water medium together with very small amounts of SLS, to load a milk thistle extract standardized at 91% in SLM content [50]. The resulting microparticles were characterized by a narrow size distribution with mean value of 4.4 μm, and extract EE of about 85%. The water-soluble microencapsulated powder dramatically improved both in vitro dissolution and permeation rates, suggesting a higher SLM bioavailability after oral administration.

An alternative methodology for the preparation of SIL-based nanodispersion with enhanced dissolution rate was performed by Cui et al., using a microchannel antisolvent precipitation combined with spray-drying [51]. That work was the first reporting SIL-loaded nanodispersion produced via microfluidics, enabling efficient control over the particle size, homogeneity, and drug release performance. The antisolvent precipitation with a syringe pump (APSP) was employed to produce SIL-loaded NPs in order to improve the bioavailability of the hydrophobic cargo [52].

The anticancer efficacy of SIL nanosuspensions was also tested by Zheng et al. [53], by carrying in vitro assays on human prostatic carcinoma PC-3 cell line. The HPH method has been recently used to test the feasibility of SIL nanocrystals as stabilizing agent of a Pickering emulsion of glyceryl monocaprylate oil droplets in water and provide a novel formulation free of any surfactants or polymer stabilizer [54]. The authors obtained flat spherical drug nanocrystals with mean particle size of 300 nm for homogenization pressure as high as 100 MPa. Scanning electron micrographs supported a core-shell microstructure of the emulsion characterized by a core of oil saturated with SIL and a shell of SIL nanocrystals. Compared with SIL nanocrystalline suspension (SN-NCS), SIL nanocrystal self-stabilized Pickering emulsion (SN-SSPE) showed a better dissolution profile with a faster rate and more efficient dissolution. This difference was attributed to the fraction of drug dissolved in the oily phase of SN-SSPE, which could be released more easily than SN-NCS.

Yang et al. [55] applied the solution-enhanced dispersion approach by supercritical fluid (SEDS) technology using various polymeric excipients to produce SLM solid dispersions with improved dissolution and bioavailability of the active ingredient tested in rodents.

### 3.2. Complexes with Cyclodextrins and Phospholipids

Natural cyclodextrins (CDs) are widely used in pharmaceuticals, drug delivery systems, cosmetics, food technology and the chemical industries. They can be found in commercially available medications, including tablets, eye drops, and ointments (see ref. [56] for a recent review on the subject). Formulations based on SLM inclusion complex with β-CD were reported by Ghosh et al. [57]. They were prepared by different methods, such as physical mixing, kneading, co-precipitation and solvent evaporation. The inclusion complex prepared by the co-precipitation method led to best results regarding the sustained drug release performance. In another investigation, a lyophilized SIL-HP-β-CD complex was prepared and evaluated in vitro by Kellici et al. [58], who performed detailed physicochemical studies on the SIL-CD interactions at the molecular level and investigated the respective bioavailability on MCF-7 cancer cells. In a different study, SLM inclusion complexes with, respectively, HP-β-CD and RAMEB were developed in order to improve SLM anti-fibrotic activity at a lower therapeutical dose of 50 mg/kg, by increasing their potential solubilization and to prevent their metabolic degradation within the GIT after oral administration, [59].

Other types of molecular complexes capable to enhance the bioavailability of the active constituents of silymarin are represented by phytophospholipid complexes known as phytosomes. The phytosome unit is a molecular complex between phospholipids and standardized polyphenolic constituents in a 1:1 or 2:1 molar ratios, whose various aspects such as interactions, structures, characterization and increased bioavailability can be found elsewhere [60]. According to several studies, SIL-phytosome proved to be more bioavailable with enhanced therapeutic activities compared to the purified molecular extracts, though most of formulations have been focused on oral and topical drug administration routes [61,62]. Besides, mixtures of phospholipid and bile salts forming PC-BS mixed micelles, were found to be good nanocarrier candidates to encapsulate SIL with high loading capacity, due to their physiological compatibility and solubilizing capacity. Yu et al. [63] prepared phospholipidic micelles mixed with BS (SPMM) to solubilize SIL in their inner hydrophobic cores. From pseudo-ternary phase diagram investigations, an optimum micellar formulation was capable to solubilize the drug up to 10 mg/mL, which was also found to double in the presence of PVP [64]. An analogous nanovehicle for the water-insoluble SIL purposely designed for parenteral application was proposed by Duan et al. [65], who first prepared a SIL-PPC equimolar complex by TFD method, which was then dissolved in anhydrous ethanol together with PPC and SDC at various molar ratios. After evaporation of the organic phase, the mixed micelles loaded with SIL were reconstituted by adding double distilled water. The optimal formulation led to a SIL loading efficiency of 14.43%, which corresponded to a drug solubility in water of 10.14 ± 0.36 mg/mL.

### 3.3. Lipid-based Formulations

A widespread formulation strategy exploits lipid-based colloidal vehicles as winning option for the delivery of SIL penalized by the poor aqueous solubility. Lipid-based formulations (incorporation of the active lipophilic component into inert lipid vehicles) are used to improve the oral bioavailability of poorly water-soluble drug compounds, which include micro- or nanoemulsions, oils, self-emulsifying formulations, surfactant dispersions, proliposomes and liposomes, solid lipid nanoparticles and lipid nanocarriers, etc. These lipid formulations can be broadly divided into two groups, namely, liquid oil-in-water (O/W) emulsions (LEs) and solid lipid nanoparticles (SLNs). The LE systems could be lipid solutions, emulsions, microemulsions and self-(nano-/micro-) emulsifying drug delivery systems. SLNs are novel lipid-based formulations which are constituted exclusively of biodegradable lipids such as highly purified triglycerides, monoglycerides, hard fats, complex glyceride mixtures or even waxes, which are solid at physiological temperature.

#### 3.3.1. Micro- and NanoEmulsions

The formulations based on both microemulsion and nanoemulsion strategies for oral/topical administration, are designed with food-acceptable or generally recognized as safe (GRAS) components to increase solubility, stability, and improve the SIL permeability. For example, O/W microemulsions were formulated to incorporate 2% w/w of SLM as a potential dermal delivery system. In particular, microemulsions containing IPM as oil phase emulsified with a 1:1 mixture of water-dispersible Labrasol^®^ and HCO-40^®^ nonionic surfactants and Transcutol^®^ as cosurfactant, enhanced drug solubility while maintaining adequate physical and chemical stability [66].

SLM-loaded lipid O/W emulsions were optimized by testing the emulsification properties of soybean lecithin as surfactant and Tween 80 as cosurfactant in combination with several food grade oils (soybean oil, castor oil, and olive oil) [67]. The authors reported that SLM was added to the lipid phase in soybean oil as 10% aqueous solution in 1 M NaOH up to an optimum drug loading of 1% w/w to produce an emulsion stable for 35 days, constituted by oil droplets with size distribution range of 0.31–1.24 μm and median diameter of 0.46 μm. The in vitro drug release behaviour was faster from the SLM-loaded lipid emulsion as compared with a SLM propylene glycol solution.

To overcome the disadvantage of the administration of large volumes of (micro)emulsions per dose, several researchers focused their studies on the design of preconcentrated organic liquid phases loaded with poorly water soluble drugs, capable of reconstituting the (micro)emulsion spontaneously once in contact with an aqueous milieu (gastric fluids after ingestion). The resulting formulation is commonly termed in the literature as a liquid self-emulsifying drug delivery system (SEDDS) [68,69]. The essential property that must be satisfied in the development of a liquid SEDDS formulation is that the drug must remain partitioned within the O/W droplets after dilution with the aqueous medium in the GIT. Otherwise, the drug could undergo an unwanted precipitation that would lead to poor bioavailability in vivo. Among the first documented studies on the application of SEDDS as potential nanocarrier to increase SIL solubilization and its oral bioavailability, we mention the work by Wu et al. [70], who optimized the multi-component oily phase, ethyl linoleate/Tween80/ethyl alcohol, to solubilize SLM. The self-emulsifying properties of that system were tested upon titration with various aqueous media until a stable O/W microemulsion was obtained. During the titration, the samples were agitated gently in order to reach equilibrium quickly. Dilution volume had no significant effect on droplet sizes (mean value of dozens nm), which were found also unaffected by the increase of drug loading up to 100 mg per 1 g of oil phase. Relative drug bioavailability after oral administration of the said lipid emulsion, improved approximately 1.88- and 48.82-fold that of SLM dissolved in PEG 400 solution and aqueous suspension, respectively.

Another optimized SEDDS was prepared using 10% GMO as the oil phase and 15% SLM, with 37.5% of a surfactant mixture of Tween 20 and HCO-50 (1:1), and 37.5% Transcutol^®^ as cosurfactant [71]. The authors evaluated also the drug solubility in various solvents at 25 °C. The O/W microemulsion was generated upon water addition until reaching a maximum water content of 95.4%. After aqueous dilution, the mean droplet size of the internal oil phase was about 67 nm. The release rate of the drug from the SEDDS measured through in vitro dissolution tests, was approximately 2.5 times higher than that from the reference commercial product (Legalon^®^). After oral administration, SLM-loaded SEDDS showed a 360% higher bioavailability compared with the reference formulation.

In a similar subsequent investigation, Li et al. [72] optimized SLM-loaded SEDDS formulations based on ethyl linoleate, Cremophor^®^ EL and ethyl alcohol, which were selected regarding the self-microemulsifying ability, solubilization ability, and reduced use of surfactant. The best combination of ingredients screened after a systematic pseudo-ternary phase diagram study, yielded a drug solubility of 130.8 mg/mL homogeneously dispersed in small O/W droplets with mean size in the range 20–30 nm and no changes were detected at 40 °C for 3 months. Both the in vitro release performance and in vivo bioavailability after oral administration of SLM from SEDDS were evaluated and compared with the commercial preparation Legalon^®^. The relative drug bioavailability of SEDDS to commercial suspension was 227%.

To overcome the side effects caused by high surfactant levels usually employed in SEDDS, Wei et al. [73] designed a supersaturable SEDDS (S-SEDDS) formulations to improve SIL oral bioavailability. It consisted of a reduced amount of surfactant in combination with HPMC added in the liquid SEDDS to induce a supersaturated state in vivo by preventing or minimizing the SIL precipitation. The authors evaluated also the SIL solubility in various oils, surfactants and cosurfactants at 25 °C. Labrafac^®^ CC showed the highest drug solubility and was selected as an oil phase for the formation of S-SEDDS. Cremophor^®^ RH 40 was chosen as a surfactant for its good emulsion-forming ability and smaller droplet size of the optimized SIL loaded emulsion (~50 nm), thanks also to the synergic effect of the compresence of Transcutol^®^ and Labrasol^®^ as cosurfactants. From in vitro studies, it was confirmed the stabilizing effect of HPMC in maintaining high SIL solution concentrations (supersaturated state). Precipitates collected after the in vitro tests from the S-SEDDS formulated with HPMC, were identified as amorphous SIL while crystalline precipitates were found when HPMC was absent in the formulation. Relative drug bioavailability after oral administration of a SIL dose of 533 mg/kg was found higher for S-SEDDS than SEDDS, i.e., same formulation without HPMC. In particular, C_max_ was 16.1 μg/mL as compared to that of the SEDDS formulation (5.68 μg/mL), while AUC of SIL from S-SEDDS was approximately 3.0-fold higher than that of SEDDS.

Stable nanoemulsions for SLM delivery containing Labrafac^®^ as an oily phase, Solutol^®^ HS 15 as surfactant, Transcutol^®^ as co-surfactant, and water as aqueous phase were developed by Adhikari et al. [74], to test the radioprotective potential of SLM-loaded nanosuspensions against γ-radiation-induced oxidative damage in human embryonic kidney cells. The formulations were designed to act as SEDDS after water addition, until the oil phase was 10–15% at the end of the dilution process. HEK cells viability upon treatment with variable concentrations of SLM-SEDDS and bare drug suspensions as control was checked prior to irradiation. Radiation-induced apoptosis was estimated by microscopic analysis and cell-cycle estimation. The proposed formulation based on SEDDS technology to improve the SLM bioavailability was found radioprotective, supporting the possibility of developing new approaches to radiation protection via colloidal dispersions of SLM.

A very stable O/W nanoemulsion consisting of SLM solubilized in nanosized oil droplets dispersed in an aqueous medium by a mixture of Tween 80 and ethanol as co-surfactant, was optimized through a systematic investigation on experimental pseudo-ternary phase diagrams [30]. Among various tested oils such as OA, IPM, Triacetin, the highest value of SLM solubilisation was observed in Sefsol^®^ 218 (~183 mg/mL), which was selected for the nanoemulsion formulation at 5% w/w in water. After oral administration of drug solubilized in the nanoemulsion, AUC and C_max_ were, respectively, 199.45 ± 56.07 μg h/mL and 31.17 ± 7.56 μg/mL, namely, 4-fold and 6-fold higher than those of the correspondent SLM aqueous suspension.

A SLM-loaded liquid nanoemulsion was formulated by Yang et al. [75], using the SPG membrane emulsification technique and then spray-dried to obtain solid state NPs. Dissolution, bioavailability, and hepatoprotective activity in vivo were assessed by comparison with a commercially available SIL-loaded product. Optimal formulation was composed by SIL, castor oil, PVP, Transcutol^®^, Tween 80, and water at the weight ratio of, respectively, 5:3:3:1.25:1.25:100. The mean sizes of the SLM-loaded nanoemulsion and NPs obtained after spray-drying were about 170 and 214 nm, respectively. The drug bioavailability after oral administration from the NPs was about 1.3-fold higher than that obtained with a commercial product (Legalon^®^).

In a more recent investigation, SLM enriched nanoemulsions were formulated using different oils such as sunflower, EVO and castor oils, respectively, [76]. SLM solubility in castor oil was 0.668 ± 0.072 mg/g whereas in EVO and sunflower oils was rather lower. However, upon addition of Tween 80 in oil (10 mg/g), drug solubility increased up to reach 1–2 mg/g in the three tested oils. Coarse emulsions were first prepared by mixing 200 g oil+SLM with 1 L of aqueous phase and then subjected to HPH, giving rise to final droplet sizes in the range 200–300 nm. Moreover, it was found that the greater the oil susceptibility to oxidation, and thus the formation of oxidation products, the greater the drug degradation incorporated into nanoemulsions.

In another recent study, Nagi et al. [77] employed a Box-Behnken statistical design (BBD) to optimize SLM-loaded nanoemulsions using Capryol 90 as oil phase capable of solubilizing SLM up to 40 mg/mL, in terms of various factors such as processing pressure and number of cycles of HPH technique and amount of surfactant/cosurfactant mixture. The non-ionic hydrophilic surfactant Solutol^®^ HS 15 was selected due to its high capacity to solubilize hydrophobic drugs and low toxicity (LD50 > 20 mg/kg) [78], and used in combination with Transcutol^®^ selected as cosurfactant on the basis of the good miscibility with Capryol 90 and Solutol^®^ HS 15. The optimal nanoemulsion satisfying the criteria of low droplet size and low PDI, necessary for high drug release potential and drug permeation through the GIT membrane, was characterized by nanodroplets of about 50 nm (PDI: 0.45) and zeta potential −31.49 mV. The values of AUC and C_max_ of that nanoemulsion formulation after oral administration were found equal to 28.69 ± 3.28 μg h/mL and 3.25 ± 0.48 μg/mL, respectively, i.e., 1.9-fold and 2.7-fold higher than those of a marketed drug suspension.

In a recent study [79], it was reported that a commercial SLM extract could be completely solubilized at the dosage of 40 mg/mL in a nanoemulsion formulated using 2.5 g of Labrasol^®^ (20%) as the oil phase and 2.5 g of Cremophor^®^ EL/Labrafil^®^ as the surfactant/cosurfactant mixture in a 1.5: 1 ratio, the remaining mass being deionized water (60%). Besides, the authors determined selectively the solubility of the main constituents identified in SLM extract, such as, TXF, SILcr, SIL and isoSIL, in various oils, surfactants, and cosurfactants to ascertain the appropriate components of the nanoemulsions. The SLM extract loaded within O/W nanodroplets showed excellent physical and chemical stability, as the size (30–40 nm) and PDI (0.114–0.179) were unaffected and no degradation of active constituents was recorded over 40 days of observation. In vitro permeation studies were performed to determine the suitability of the prepared nanoemulsion for oral delivery.

#### 3.3.2. Liposomes

Liposomes are hollow spherical NPs with a closed shell of a lipid membrane (mono- or multi-layer), inside of which an aqueous solution can be encapsulated. These supramolecular aggregates owe their success as carriers of therapeutic drugs for many advantages including the capability to encapsulate both hydrophilic and lipophilic drugs, having targeting and controlled release properties, cell affinity, tissue compatibility, reduced drug toxicity and improved drug stability [80]. Moreover, liposomal systems are known to find an immediate access to the reticulo-endothelial system (RES) rich sites like liver and spleen, and this self-targeted nature of liposomal carriers can be exploited well for drug distribution to hepatic site. During the researches, the conventional structures of the liposomes have been subjected to several changes, which have brought out a series of new type liposomes, such as long-circulating stealth liposomes, stimuli-responsive liposomes, cationic liposomes and ligand-targeted liposomes [81]. Liposomes can be prepared using a wide range of methods, such as thin-film dispersion (TFD), reversed-phase evaporation (RPE), alcohol injection, and spray-freeze-drying. Other strategies comprise proliposomal formulation, with the use of a cryoprotectant and high process temperature, and supercritical fluid of carbon dioxide method (SCF-CO_2_), which is a flexible and environmental-friendly technique by which particle sizes and shapes can be controlled by tuning the experimental conditions (temperature and pressure).

One of the first liposomal formulations of SLM reported in literature dates back to the early 2000s. The study by Maheshwari et al. [82], focused on the development of factors such as the drug to lipid ratio, the proportion of CHOL and presence of the charge inducer DCP in the optimization of the formulation prepared by ethanol injection. The highest drug entrapment of about 95% was achieved in the formulation with SLM/PC/CHOL/DCP ratio of 2:10:2:1. The obtained size range of SLM-loaded liposomes (56−1270 nm with median diameter of 390 nm) was suitable for i.v. administration for hepatoprotective studies in mice. A drug leakage of about 40% was observed in 28 days as well as evident aggregation phenomena recorded after 3 weeks of the preparation.

To overcome instability problems that commonly occur in the GIT and improve the poor aqueous solubility of SLM, El-Samaligy et al. [83], investigated the feasibility of encapsulating the drug in a liposomal dosage-form for buccal administration via spray. Liposomes were prepared by RPE method using a base lipid mixture of soybean lecithin and CHOL in a 9:1 optimized molar ratio. In addition to the basic liposome ingredients, various additives were gradually introduced, such as positively (SA) or negatively (DCP) charge inducers and non-ionic surfactants (Tween 20 or Tween 80). At the end of a multifactorial screening, an optimal composition for hybrid liposomes was derived as lecithin/CHOL/SA/Tween 20 at 9:1:1:0.5 molar ratios, which warranted both best SLM EE of about 69% and high in vitro absorption and permeation performances [84].

In order to improve the stability of liposomal nanocarriers and enhance the SLM encapsulation, Xiao et al. [85] adopted the strategy of “proliposomes”, which are defined as dry, solid particles that form a reconstituted liposomal suspension when put in contact with water [86]. SLM-proliposomes were prepared by film-deposition using mannitol as carrier and a mixture of methanol and chloroform (2:1 v/v) as the apolar medium to dissolve both drug and phospholipids. The content of SLM in the proliposomes was 9.73% (w/w). In the reconstituted liposomal suspension, the mean particle size was about 200 nm while a mean value for SLM EE was about 93%. After oral administration in beagle dogs of drug entrapped in the reconstituted liposome suspensions (drug equivalent to 7.7 mg kg^−1^), the pharmacokinetic parameters AUC and C_max_ were, respectively, 2.46 ± 0.58 μg h/mL and 0.47 ± 0.13 μg/mL.

The proliposome nanotechnology was also exploited to improve the water solubility and bioavailability of 2,3-dehydrosilymarin, an oxidized form of SLM characterized by significantly greater antioxidant and anti-cancer activity than the reduced precursor [87]. Hence, the 2,3-dehydrosilymarin-loaded proliposome powder was produced by the TFD-freeze drying method obtaining a polyphase dispersed system composed of phospholipids, CHOL, IPM and SC, with optimal drug-lipid ratio set to 1:3 [88]. The correspondent drug content after reconstitution with water was 25.00 ± 5.93 μg/mL, yielding EE of about 82%, which was predominately dependent on the drug/phospholipid and SC/phospholipid ratios. The improved oral absorption in rabbits was ascribed to the relatively small size of liposomes distributed in the range 7–50 nm and average diameter of about 16 nm. The AUC and C_max_ were approximately 2.29-fold (12.77 ± 1.39 μg h/mL) and 4.96-fold (2.83 μg/mL) higher than those of the simple 2,3-dehydrosilymarin suspension.

The strategy of ligand-functionalized liposomes as nanocarriers for SLM was adopted by Elmowafy et al. [89] to design PEGylated liposomes decorated with the hepatic targeting ligand Sito-G (a carbohydrate epitope). The surface modified liposomes prepared by TFD method, contained fixed 2:1 molar ratios of HSPC and CHOL with or without PEG as well as a varying concentration of Sito-G to check its optimal amount for improving the targeted delivery of the encapsulated SLM to hepatic target cells. The combination of relatively high melting point HSPC with a high percentage of CHOL provided liposomes with very rigid lipid bilayers. The obtained multilamellar vesicles were exposed to several freeze-thaw cycles in which the liposomes were frozen in liquid nitrogen for 2 min and defrosted in warm water bath for 2 min. The submicron-sized liposomes were prepared by using a pressurized extruder with two polycarbonate membrane filters with pore size of 100 nm. The final lipid concentration in the liposome formulations was 4 mg/mL. The particle size distribution of the liposomes with mean values in the range 145–168 nm showed very good homogeneity (PDI 0.15–0.3). An acceptable drug EE was recorded for all tested formulation (about 60% in average), although high contents of Sito-G were found detrimental to the stability of liposomal membrane, thus leading to a decrease in the EE. A systematic investigation of the in vitro release profiles performed with the dialysis method (37 °C in HEPES buffer, pH = 7.4), indicated that PEGylated liposomes exhibited a sustained SLM release as compared to conventional liposomes. On the other hands, PEGylation of liposomes equipped with Sito-G manifested a reduced SLM cellular uptake with HepG2 cells compared to non-PEGylated nanocarriers.

A more recent development of using ligand-functionalized liposomes to improve the SIL bioavailability is reported in the work published by Ochi et al. [90], who demonstrated a synergic effect on the liver cancer cell line HepG2, provided by the co-encapsulation into PEGylated nano-liposomes of SIL and GA. The liposomal suspensions, prepared by the TFD method followed by sonication, were formulated with SIL and GA at a 1.74:1 molar ratio together with a mixture of DPPC, CHOL, and mPEG2000-DSPE at a specified molar ratio. Scanning electron microscopy analyses showed that the co-encapsulated nanoliposomes had a mean diameter of 43 nm while the zeta potential was −23.25 mV, sufficient to inhibit liposome aggregation.

Besides enhancing drug bioavailability, the liposomal formulation developed by Kumar et al. [91], was designed to promote the regeneration of hepatocytes and to prevent inflammation in liver. The SLM EE of liposomes prepared with the TFD technique, was found to be maximum (55%) for formulation containing SPC and CHOL at molar ratio 6:1. The optimal liposomal formulation yielded a 3.5-fold higher SLM bioavailability (AUC 0.500 ± 0.023 μg h/mL) than the correspondent drug suspension. Likewise, C_max_ was found 5.25-fold (0.716 ± 0.043 μg/mL) higher than the drug suspension. Analysis of in vivo studies suggested that SLM encapsulated in said liposomal carriers might have targeted inflammatory cells resulted in increased anti-inflammatory activity.

A slightly different approach was explored by Angelico et al. [92] in the preparation of liposomes loaded with SIL-phytosome rather than using the commercial purified SIL extract in the formulation. The addition of lecithin in the starting lipid film based on SIL-phytosome such that the final phospholipid/SIL ratio reached 6:1, yielded stable phytoliposomes with the suitable surface charge and average dimensions in view of a potential parenteral i.v. use, where the nanoparticle size is a critical parameter to be controlled. The cellular uptake of SIL encapsulated into phytoliposomes and its antiviral activity were also tested in vitro with Huh7.5 cells [93]. The data clearly demonstrated that the cell absorption was 2.4-fold more efficiently than free SIL, and 300-fold more potent pharmacological activity. It is worth noting that the phytoliposomes were able to reduce the hepatitis C virus infection by inhibiting the entry of viral particles into cells.

Methods that use SCF-CO_2_ technique have been also employed for the preparation of SLM-loaded liposomes. In particular, the solution-enhanced dispersion by supercritical fluids (SEDS) has been adopted to produce liposomes constituted by HSPC and SGC as base lipid ingredients, [94]. The optimized product provided better performances than liposomes prepared with more conventional methods such as RPE or TFD. Based on the analysis of drug release profiles in vitro, the novel formulation improved the SLM solubility. In vivo tests showed an improved oral bioavailability of drug administered in liposomes containing BS (AUC 18.406 ± 1.481 μg h/mL; C_max_ 1.296 ± 0.137 μg/mL) compared to an aqueous drug suspension or a commercial product.

A recent study reported on the effect exerted by SLM contents and type of BS on the drug EE in BS-liposomes (bilosomes) prepared with TFD method [95]. The obtained vesicle dispersions were characterized by highly negatively charged zeta-potential values compared to an analogous preparation formulated with CHOL in place of BS. Considering particle dimensions, bilosomes containing SC exhibited the largest particle diameter (595.1 ± 98.48 nm) among all BS investigated. Among all the screened formulations, the optimum composition in terms of highest SLM EE (84.54%) corresponded to a lipid molar ratio 4:1 for SPC/SC system. In vitro release studies revealed biphasic pattern of all formulations while in vivo investigations revealed that bilosomes showed a pronounced effect in retaining the hepatoprotective as well as oxidative stress biomarkers to their normal levels against CCl_4_ induced hepatotoxicity.

Liquid crystalline (LC) cubosomes coupled to P407 have been also formulated to enhance the oral bioavailability of SLM [96]. The LC matrix system was prepared by a melting/congealing method with GMO to P407 ratio of 100:12 at which cubic LC phases formed upon hydration. The amounts of drug dispersed as amorphous state in the matrices were fixed within the range 2–8%. SLM entrapped into the GMO-P407 LC system, manifested a 3.5-fold increase in bioavailability after oral administration as compared with a commercial drug formulation. Finally, for an applicability of SLM in the treatment of atopic dermatitis (AD), a novel formulation was designed by using pluronic-lecithin organogels, owing to their biphasic composition and versatility as transdermal and topical DDs [97]. The tested formulations contained 20% oil phase (lecithin/IPM) and 80% aqueous phase (pluronic). The high penetrating ability and hydration effect of the organogel base, provided a significant improvement in the signs and symptoms of AD patients.

#### 3.3.3. Solid-Lipid Nanoparticles (SLNs), Nanostructured Lipid Carriers (NLCs)

Solid lipid nanoparticles (SLNs) are the first generation of lipid-based nanocarriers that are formulated from lipids, which are solid in the body temperature and stabilized by emulsifiers [98]. Emulsomes are special case of SLNs, considered as the solid state version of common uninamellar and multilamellar lipid vesicles, i.e., NPs with an internal solid fat core surrounded by one or more phospholipid layers [99]. For an exhaustive discussion about pros and cons occurring in the use of this type of lipid nanocarriers see, e.g., a recent review by Ghasemiyeh et al. [100]. Differently from matrices having either solid (SLNs) or liquid lipids (LEs) as core composition, the nanostructured lipid carriers (NLCs), which belong to the second generation of lipid NPs, represent hybrid formulations prepared by blending solid lipids and liquid lipids, thus resulting in a less ordered inner structure [101].

The application of SLNs drug delivery strategy was pursued by Xu et al. [102] who first prepared a SIL-PC complex at a mass ratio of 1:3 by TFD method, which was subsequently dissolved in anhydrous ethanol together with a water-soluble derivative of vitamin E (TPGS) and PC at ratios 4:1:20, w/w/w. After ultrasonication and further solvent evaporation, the resultant lipid film was hydrated in distilled water leading to a lipid suspension, which was sonicated until a translucent colloidal dispersion was obtained. The aim was to test the anti-metastatic effect of SIL in combination with TPGS co-entrapped in SLNs, to suppress effectively the metastasis of breast cancer both in vitro and in vivo. The average particle size was about 45 nm with a zeta potential value of 2.78 mV and the SIL entrapment efficiency (EE) was approximately 99%. The in vivo results supported the potential use of SIL-TPGS loaded SLNs as an anti-metastasis agent capable to inhibit the invasive and metastatic activities of breast cancer cells instead of acting a cytotoxic effect against them.

Furthermore, in an interesting work by Shangguan and coworkers [103], it was reported the performance in animal models of both SLM-loaded SLNs and SLM-loaded NLCs, by comparing their oral bioavailability with that of their lipolysate counterparts and fast-release formulations. The goal was to determine whether and to what extent the integral lipid NPs contribute to the overall bioavailability of SLM selected as a poorly water-soluble model drug.

In an earlier study, SLM-loaded SLNs were developed using the emulsifiers Compritol 888 ATO, soybean lecithin and P188 and both hot and cold variants of the HPH method were compared each other to check differences in drug incorporation modes and release mechanisms [104]. From the analysis of results obtained by centrifugal ultrafiltration method, the SLM EE in the cold preparation reached 87% with a fraction of adsorbed drug of about 8%, while for hot homogenization the EE was 43% and the fraction adsorbed was 54%. The in vitro tests carried out by reverse dialysis bag technique at pH 7.4, showed a prolonged drug release for SLM-SLNs produced by cold homogenization. In vivo studies after oral administration of the cold preparation showed high drug levels and long residual time in the plasma and liver after oral administration of the cold preparation compared to SLM aqueous suspension.

Experimental evidences of the anti-hepatotoxic property of SLM encapsulated in SLNs were reported in Cengiz et al. [105], who tested their formulations against the liver damage induced in model animals by the administration of a combination of the hepatotoxin D-GaIN and TNF-α. The solid lipid nanosuspensions based on Compritol and Tween 80, were prepared through the hot homogenization technique. The resultant colloidal SLM-loaded dispersions were characterized by particle sizes varying in the range 165–200 nm with zeta potential of −26.5 mV. Both in vitro and in vivo tests demonstrated that SLM-loaded SLNs were more effective than control in curing liver damage, mainly owing to the slow and regular drug release.

To reduce the disadvantage of nanocarrier elimination from blood circulation by the reticuloendothelial system (RES), Zhang et al. [106] proposed a novel SLNs formulation based on SA covalently grafted with PEG 1000 (Brij 78) as stealth agent to form a hydrophilic steric barrier around the lipid NPs. SIL-loaded stealth SLNs were prepared according to the EES method, which consisted in the preliminary preparation of a lipid emulsion, obtained by adding dropwise an organic phase of SA and SIL to an aqueous phase containing the non-ionic surfactant Brij 78. After solvent removal by evaporation, the resulting emulsion system was quickly added to a given volume of cold water to obtain a final suspension of SIL-loaded stealth SLNs. The authors observed that the rate of addition of organic phase to the aqueous phase was the crucial step in preparing the emulsion. A systematic investigation was carried out in order to optimize the drug EE and the homogeneity of particle size distributions. The mean particle size of the optimized formulation was 179 nm (PDI 0.168) and zeta potential of −25 mV, which represented quality values of good stability. In vitro assays by dialysis method revealed a very slow drug release, a property considered beneficial for SIL bioavailability after oral administration.

In an analogous application of the EES method to prepare SLM-loaded SLNs, a lipid phase composed of SA and Capryol 90 was emulsified in aqueous phase containing Brij S20, and subsequently added to cold water to induce the formation of lipid NPs. The mean particle size, zeta potential and EE were found correspondent to, respectively, about 214 nm, −32 mV and 92% [107]. The ability of NLCs to prevent the burst release and gastric deterioration of SLM has been documented by in vitro release studies. Moreover, permeation studies indicated that the formulations were successful in enhancing the cellular uptake of SLM and that the transport of the NLCs across the cell monolayer was energy-dependent and mediated by clathrin- and caveolae (or lipid raft)-related routes.

With the aim of improving the drug loading in SLNs and therefore increasing its oral bioavailability, Ma et al. pursued the strategy to encapsulate fatty acid (FA)-SIL conjugates into lipid nanovehicles and compare the oral fate with bare SIL [108,109]. Three types of FA derivatives characterized, respectively, by 6C-, 12C- and 18C-long chains attached at level of 7-OH position of SIL (see Figure 1), were encapsulated in SLNs prepared with hot HPH method using Precirol^®^ ATO 5 as lipid matrix and Tween 80 as the emulsifier. The mean particle diameter and zeta potential determined for the various SLN formulations with EE 100% were, respectively, in the range 102–140 nm and between −11.1 and −13.5 mV. Lipolysis studies conducted in SIF indicated that SLNs loaded with each of the three types of FA-SIL underwent complete degradation within 70 min, showing lower rates for SIL conjugates with longer chains. Based on the pharmacokinetic results, the authors argued that the lymphatic route could play a leading role in the transport of integral SLNs. Notably, taking together all the results obtained from both in vitro and in vivo observations, the authors were able to deduce the fate after oral adsorption of FA-SIL conjugates encapsulated in SLNs and the underlying mechanisms of enhanced bioavailability.

Thanks to the good biocompatibility and biodegradability, the emulsomes nanoencapsulation technology has been recently applied to provide particle stability, high EE, and sustained SIL release, [110]. The preparation followed the TFD method with the variant of choosing a solid lipid matrix at r.t. such as triglycerides composed of natural unbranched fatty acids or glycerol triesters of saturated fatty acids (e.g., trilaurin), in place of biocompatible oils as internal phase in which SIL has to be dissolved. Other ingredients were CHOL and Tween 80 mixed together with trilaurin and PC in chloroform solution. The film obtained after solvent removal was hydrated and homogenized by ultrasonication to obtain SIL-emulsomes, which were characterized by average particle diameter of about 364 nm and zeta potential of −34 mV. SIL-emulsomes exhibited sustained release in vitro and better pharmacokinetic parameters in vivo compared with SIL solution as control.

Regarding the ability of NLCs to efficiently entrap SIL and thus increase its bioavailability and consequent therapeutic activity, it can be mentioned the work published by Jia et al. [111], who exploited the potential of this formulation strategy for i.v. SIL delivery. For the preparation of SIL-NLCs with the ESE method at a high temperature followed by solidification at a low temperature, GMS was used as solid-lipid ingredient while MCT was selected as liquid-lipid material. First, a nanoemulsion was obtained by adding dropwise a hot alcoholic solution of SIL, lipids (GMS and MCT) and lecithin, to an aqueous phase containing 1.5% Pluronic F68 under mechanical stirring. Then, the nanoemulsion was quickly dispersed into cold distilled to promote the SIL-loaded NLC dispersion. The mean particle size was about 230 nm and zeta potential −20.7 mV. It was also observed that the drug EE and drug loading increased from 72.31 to 96.87% and from 3.63 to 4.84%, respectively, with the increase of MCT% from 0 to 30 wt% [112]. From in vitro studies, a burst drug release was detected at the initial stage due to a fraction of SIL loaded into the liquid-lipid-enriched outer layers of NLCs. Afterwards, a more prolonged drug diffusion occurred when the fraction of SIL dispersed into the NP core was gradually released by erosion of the lipid matrix. Moreover, SIL-NLCs showed higher AUC values and a prolonged residence time in the blood circulation compared with SIL solution as control.

More recent improvements in the formulation of NLCs as SLM delivery systems considered several variants of preparation such as, e.g., the hot HPH method as described by Wu et al. [113]. The resulting formulations, based on GDS and OA as the solid and liquid lipids, respectively, and lecithin and Tween-80 as the emulsifiers, yielded SLM EE and loading capacity of about 87% and 8.32%, respectively. The average particle size was ~80 nm with mean zeta potential of −65 mV. The enhanced bioavailability of SLM-NLCs (2.54- and 3.10-fold that of commercial Legalon and solid dispersion pellets, respectively) was interpreted in terms of facilitated absorption by the lipid-based DDs. Another investigation used GMS, CP and SA as solid lipids and CA and OA as liquid lipids [114]. Various mixtures of solid/liquid lipids preloaded with SLM alcoholic solution, were heated and emulsified in aqueous phase containing Tween 80. The prepared pre-emulsions were ultrasonicated and then refrigerated to favour lipid crystallization. The final separation of the resulting NPs was performed by ultra-centrifugation. All the formulations were analyzed according to the Box-Behnken factorial design. Organ distribution studies revealed that SLM loaded in NLCs containing GMS and OA (particle size ~224 nm and EE ~79%) underwent selective liver uptake through the chylomicron pathway. In Iqbal et al., [115] the organic phase consisted of solid (Geleol^®^)/liquid (Sefsol^®^ 218) mixed lipids dissolved in ethanol together with SLM, which was heated at 60 °C and emulsified by sonication at 70 °C to remove the organic solvent by ESE. The Central Composite Rotatable Design (CCRD) was employed for optimization of SLM-loaded NLC dispersions and the resulting best formulation (particle size ~126 nm and EE 85%) was freeze dried and incorporated in hydrogel matrix of Carbopol for epidermal tissue deposition to use SLM as chemoprevention of skin cancer. Finally, SEDDS and NLCs were employed to enhance SIL oral bioavailability at level of GIT and treat obesity-induced NAFLD [116].

### 3.4. Polymer-based Nanocarriers

In literature exists a variety of solutions designed by researchers to effectively encapsulate SIL in biocompatible and biodegradable polymeric nanosystems such as polymeric micelles, composites and solid nanodispersions [117]. Altogether they represent a very efficient strategy by which a poorly water-soluble drug can be dispersed into an inert hydrophilic polymer matrix. As the polymeric erosion progresses, the loaded drug is released in the form of very fine particles for rapid dissolution.

#### 3.4.1. Inclusion in Polymeric Matrices

The choice of the formulation method is fundamental to optimize the performance of the final product as described by Sonali et al. [118], who compared kneading, spray drying and co-precipitation techniques in the preparation of SLM-loaded solid dispersions using HPMC as a hydrophilic polymeric carrier. In vitro studies suggested the following enhancement in SLM dissolution compared to pure drug: co-precipitation (2.5 fold) > spray drying (1.9 fold) > kneading (1.5 fold).

Nguyen et al. [119] developed a high-payload supersaturating delivery system by preparing SIL-chitosan NPs from a drug-polysaccharide complexation (nanoplex). At the optimal pH and chitosan-to-SIL charge ratio, the size and zeta potential of nanoplex were, respectively, 243 nm and 21 mV, while the complexation efficiency and yield were obtained in the range 83–87% and 55–63%, respectively. The nanoplex stability after either short or long-term storage and prolonged supersaturation in the presence of HPMC, demonstrated its feasibility as a new strategy for improving SIL bioavailability.

Another formulation of SLM (containing 33% of SIL) loaded NPs with particle size of ~ 100 nm, was proposed by Zhao et al. [120] by ESE technique and freeze-drying method, using P188 as polymer. Absorption of SLM NPs in the organs was significantly higher than that of drug suspension and very high SLM concentrations were observed in the liver with a long retention time.

NPs formulated by loading SIL into the cationic copolymer Eudragit, were prepared by nanoprecipitation technique using PVA as a stabilizer, to investigate their anticancer efficacy in oral carcinoma (KB) cells [121] and their ability to reverse the fibrosis-induced cholestasis in rats [122].

Another recent investigation reports the formulation of Eudragit-based SLM-loaded NPs, prepared by nanoprecipitation technique utilizing different PVA concentrations at various organic/aqueous phase ratios. The formulation of suitable particle size of ~ 85 nm, characterized by EE of 83.45% and in vitro 100% drug release after 12 h, was selected for in vivo hepatoprotective activity and toxicity studies [123].

One of the recently published research reports a formulation composed by SLM-PVP-PEG (0.25:1.5:1.5, on weight basis) in form of polymeric composite, which promotes an increase in SLM solubility of more than 24 mg/mL and an excellent dissolution profile [124]. The enhancement in drug solubility and dissolution has been attributed to a better SLM wetting driven by the hydrophilic polymers, complete conversion of the crystalline components into the amorphous state and molecular level homogeneousness of SLM, PVP and PEG in the resulting polymeric composite.

#### 3.4.2. Dendrimers and Polymeric NPs

Dendritic macromolecules such as PAMAM are hyper branched biocompatible polymers, widely employed as nanocarriers for the encapsulation and delivery of compounds either by supramolecular complexation (encapsulated drug) or covalent attachment (conjugated drug) [117]. Each repetitive branching represents a new dendrimer generation (G), which determines the molecular weight, size and number of primary amine sites.

In particular, both amine-terminated full generation (G2 and G3) and ester-terminated half-generation (G1.5 and G2.5) PAMAM dendrimers, were tested for their potential use as SIL solubility enhancers [125,126]. Low molecular weight drugs such as SIL, may be encapsulated into the inner core of PAMAM dendrimers or interact with their positively charged surface groups. The estimated number of SIL molecules incorporated into dendrimers ranged between 20 and 32 for G2 and G3 while for G1.5 and G2.5 it was comprised between 4 and 6. The analysis of pharmacokinetic experiments and oral bioavailability data demonstrated that drug-dendrimer complex could improve the oral absorption of SIL.

One of the open question is the mechanism whereby PAMAM dendrimers increase the small intestinal absorption of the drug. Probably more than one mechanism, such as the effect of paracellular transport of drug-dendrimer complexes through the epithelium, improved contact with the epithelium itself and higher absorption through the endocytosis process, could contribute to the greater oral bioavailability of SIL vehicled by PAMAM dendrimers.

A recent investigation reported also the use of partially PEGylated PAMAM-G4 dendrimers as solubility enhancers and supramolecular carriers for SIL, using two intermediate PEG chains sizes and substitution percentages close to 50% [127]. It has been shown that PEGylation with 2.0 kDa PEG chains provided high drug solubility and extended drug release time compared to analogous dendritic nanovehicles decorated with shorter PEG chain (0.55 kDa). Moreover, 2D-NOESY experiments revealed that the preferential binding sites for SIL association were found at level of both internal branches and external PEG chains.

Alternative nanocarriers are represented by the polymeric micelles, used as a multifunctional drug delivery platform to improve the solubility, stability, and delivery efficiency of poorly soluble drugs. SLM solubilization into polymeric micelles was achieved using an amphiphilic derivative of the carboxymethylchitosan (CMCHS) synthetized by Sui et al. [128]. Polymer amount of 10 mg/mL improved the concentration of solubilized SLM 13 times more than that of pure drug dissolved in water. SLM-loaded polymeric aggregates were found much bigger and polydisperse than blank micelles. In vitro tests showed a slow SLM release from micellar solution, lasted up to 40 h.

Polymeric PLGA NPs, prepared by the single ESE technique, have been also used to solubilize SLM isolated from the seeds of some native milk thistle ecotypes of the Egyptian Delta as described in [129]. Then, the SLM-loaded PLGA NPs were encapsulated into a calcium-crosslinked alginate matrix to obtain a series of biodegradable pH-responsive hydrogel microparticles. The developed particles showed promising biodegradability and sustained SLM release profiles, as well as improving overall drug dissolution.

PLGA polymeric nanocarriers were also used by Snima et al. [130] to produce SLM NPs through the ESE method. The particle hydrodynamic diameter was ~ 220 nm and SLM EE was 60%. In vitro tests showed a slow and sustained drug release profile at physiological conditions. Moreover, the NPs with desired drug release kinetics, were capable of delivering SLM into prostate cancer cells to induce differential anticancer effect.

To reduce the entrapment difference observed for the active components of a plant extract due to their different physicochemical properties, some researchers have developed the concept of synchronous encapsulation of multiple components of SLM into PLGA NPs. Indeed, due to the complexity of herbal medicine compositions, it is desirable to maintain the mass ratios of the multiple components in dosage forms and to achieve synchronous release both in vitro and in vivo because the pharmacological effect would highly depend on such ratios [18].

Therefore, a SLM extract with known composition of its active components (27.40% SIL, 16.58% SILcr, 8.35% isoSIL, 4.02% SILdi and 2.57% TXF by HPLC), was efficiently encapsulated into PLGA NPs with different polymerization degrees and using PVA in the aqueous phase as a stabilizer [131].

The weight ratio of SLM:PLGA had a significant effect on the overall entrapment, especially for the most hydrophilic compound among all the components (TXF). After an initial screening of experimental variables, it was found that the EE of the relatively more hydrophilic components TXF, SILcr and SILdi was always much lower than those of SIL and isoSIL. The optimized conditions to prepare PLGA NPs with the ESE method, enabling the synchronous encapsulation of all the active compounds, were observed in correspondence of an increase of both ionic strength and pH, and PVA aqueous solutions saturated with SLM.

Analogously, SLM-loaded PCL NPs formulated by ESE technique, showed sustained release, improvement in bioavailability as well as enhancement of liver protection following i.v. administration [132]. Particle sizes and EE were 130–430 nm and 91–95%, respectively, depending on polymer concentration. Increase in polymer content led also to delayed drug release due to increase in particle dimensions.

Both in vitro and in vivo tests proved the potentially beneficial use of the developed SLM NP formulation in the treatment of cirrhosis and fibrosis diseases.

Among various methods used in the preparation of chitosan NPs, namely, microemulsion, ESD, polyelectrolyte complexation and ionic gelation, the latter was adopted by Pooja et al. [133], to prepare chitosan-TPP NPs for SIL oral delivery. The optimal formulation was characterized by EE of about 83% with particle size of 264 nm (PDI < 0.3) and positive zeta potential (37.4 mV). SIL was amorphous and homogeneously dispersed in the polymeric matrix and no chemical interaction between drug and chitosan was deduced from the FTIR spectra analysis. Cytotoxicity studies revealed that SIL entrapped in chitosan NPs was more effective as anticancer agent than free drug.

Guhagarkar et al. [134] reported the design of polyethylene sebacate (PES) NPs functionalized with the polysaccharide pullulan (PUL) as hepatic targeting agent. SLM was entrapped into the NPs prepared by nanoprecipitation method giving rise to final the PES-SLM-PUL biodegradable polymeric nanocarrier. The EE was about 43% with mean particle size of 283 nm.

The liver protection activity was ascertained in a model of induced hepatotoxicity in rats, by detecting the reduction in levels of serum transaminase. Histopathological evaluation of liver tissues also confirmed the enhanced hepatoprotection upon oral administration of PES-SLM-PUL NPs.

Ma et al. [135] used the *Bletilla striata* polysaccharide (BSP) modified with SA to encapsulate SLM into self-assembled spherical NPs with a mean diameter of 200 nm. Compared to the drug suspension, the developed formulation with SLM EE of about 80%, was able to improve cytotoxicity and cell uptake in HepG2 cell lines in vitro.

In another biopolymer-based treatment, the SLM incorporation in NPs composed by water soluble chitosan and PGA, a natural anionic peptide produced by several *Bacillus* species, has been clearly revealed to be an effective approach for improving the drug solubility and its antimicrobial activity [136]. Hence, it was demonstrated that biodegradable films containing SLM NPs could efficiently control the growth of food microorganisms.

The in vivo hepatoprotective activity of SLM entrapped into chitosan NPs (SLM-NPs) by ionotropic gelation method, and the anti-inflammatory effect of inulin nanoparticles (IN-NPs) synthesized using the emulsion method, were evaluated singly or in combination against the hepatotoxicity induced by the mycotoxin DON [137]. Biochemical results showed that the combined treatment with SLM-NPs plus IN-NPs overcame significantly the toxicity of DON in the liver.

### 3.5. Nanostructured Materials Based on Inorganic Compounds

Inorganic nanomaterials are currently considered as powerful and very efficient drug carriers due to their versatile nanostructure, functional properties and controlled drug release behaviors. Moreover, they can exhibit excellent biocompatibility, biodegradation, in vivo stability, low cytotoxicity and nonimmunogenic profiles, thereby making these nanovectors an ideal candidate for oral and parenteral drug delivery [138]. One of the first applications of these inorganic nanomaterials as carriers of a water-soluble form of SIL (molecular complex with the amino-sugar meglumine), was provided by Cao et al. by engineering hollow-type mesoporous silica NPs (HMSNs) [139]. They were prepared in a W/O microemulsion formulated with nonylphenol 10 as surfactant and subjected to ultrasound irradiation to generate micropores and microchannels within the silica spheres. The obtained HMSNs with particle sizes within the range of 50–100 nm and large specific surface area (386 m^2^/g), were able to load silybin meglumine with EE of about 69%. Correlations between in vitro dissolution rates and in vivo absorption rates were also reported for HMSNs. Other studies reported the synthesis of monodispersed porous silica NPs (PSNs) by applying an ultrasonic corrosion method to create regular nanometer-sized pores with sodium carbonate solution [140,141]. SIL-loaded PSNs were obtained by immersing the porous NPs in SIL ethanol solution and stirring for 24 h, and after various treatments the entity of drug entrapment in PSN was quantified in about 69%. The mean hydrodynamic particle diameter was ~57 nm, with unimodal and narrow size distribution. In vivo studies indicated multistage release pattern after oral administration of SIL-loaded PSNs, where an initial delay in the rate of drug absorption was followed by an enhanced extent of absorption with a high plasma concentration maintained even up to 72 h. Those results suggested that the PSNs could be used as a promising SIL nanovectors for sustained-release systems, satisfying the need for prolonged treatment after oral administration.

Commercially available carboxylated multiwalled carbon nanotubes (COOH-MWCNTs) [142], have been employed by Tan et al. [143] to covalently conjugate SIL for the advanced drug delivery in therapeutic applications. Drug release from the carbon nanotubes showed a sustained and pH dependent behaviour while a significant in vitro cytotoxicity was expressed against two human cancer cell lines at lower concentrations of 1.56–6.25 μg/mL when compared to free drug alone.

Other types of inorganic NPs, such as amorphous calcium phosphate (ACP) nanospheres and crystalline hydroxyapatite (HAP) nanorods, which enjoy favourable chemical properties similar to the inorganic constituents of natural bone tissue, were exploited by Chen et al. [144] to encapsulate SIL. Drug loading into ACP and HAP was achieved by immersion of both types of NPs, previously synthesized by polymeric micelle-templated technique, in SIL-containing ethanol solutions. Both ACP nanospheres and HAP nanorods manifested relatively high drug loading capacity of 900 and 825 mg g^−1^, respectively. The drug release in both simulated intestinal (SIF) and gastric (SGF) fluids of ACP and HAP delivery systems, exhibited a rapid SIL release at the early stage (< 2 h), followed by a slow and sustained release in a period of about 17 h.

Furthermore, a novel micelle-templated synthesis of porous calcium phosphate microparticles was developed by Zhu et al. [145], to improve the delivery of the poorly water-soluble SIL in vitro and in vivo. In particular, a mixed micellar system composed by PVP/SC/phospholipid in suitable mass ratios, was adopted innovatively as a new type of template to fabricate the calcium phosphate micron-sized carriers. The drug encapsulation was proven to be incorporated into the porous structure of microparticles after the removal of the micellar template, thus leading to prolonged release in vitro and enhanced absorption in vivo. An excellent linear relationship was obtained between in vitro dissolution and in vivo absorption data, recorded in two media at different pH. This correlation could suggest the possibility to predict in vivo pharmacokinetic behavior through the observed in vitro release profiles.

An example of nanohybrid materials for the advanced delivery of SIL in therapeutic applications is represented by SIL-loaded magnetite NPs (Fe_3_O_4_) modified with PLGA-PEG copolymers as reported by Ebrahimnezhad et al. [146], to investigate their inhibitory effect on telomerase expression in T47D human breast cancer cell line. The PLGA-PEG-Fe_3_O_4_ NPs with a SIL loading capacity of about 76%, besides being biocompatible, possessed the advantage of being directed into the target tissue by the action of an external magnetic field [147]. The cytotoxic effect on the T47D cell line increased with increasing the concentration of SIL-loaded NPs, demonstrating the feasibility of this nanodrug in down-regulation of Telomerase gene expression in cancer cells.

In a more recent formulation, Fe_3_O_4_ NPs were coated with chitosan by the coprecipitation method and then loaded with SLM [148]. The capability to act simultaneously as drug nanocarrier and magnetic resonance imaging (MRI) contrast agent was tested through various methods and techniques. The zeta potential of bare magnetite NPs in aqueous dispersion changed from negative (−24.2 mV) to positive (+31.6 mV) values upon coating with chitosan, demonstrating the presence of terminal amino groups on the particle surface. The average particle size was about 18 nm with SLM EE of 95%. In vitro studies revealed a sustained drug release pattern.

Another system containing stimuli-sensitive components was reported by Fazio et al. [149], where SIL and gold nanocolloids synthesized by laser ablation were co-loaded into the PLGA-PEG copolymer in a single step procedure. The SIL-loaded PEG-PLGA-Au nanocomposite with a polymer/drug weight ratio of 50:5, was prepared by a modified emulsion-diffusion method. The hybrid NPs allowed SIL to be released in a controllable manner upon thermal activation of Au NPs incorporated in the polymer matrix, stimulated by irradiation of a red laser source of low power density (21 mW cm^−2^), partially transparent to human flesh. The localized and intensive heating of Au NPs causes the thermal expansion of the polymer, with the consequent release of drug that starts to diffuse out. As a potential application of this nanocomposite, it was suggested the use of wirelessly controlled nanowires responding to an electromagnetic field generated by a separate device. This engineering system would eliminate the tubes and cables required by other implantable devices with the risk of infections and other complications, and activate the release of the drug near areas of the body that are often difficult to reach.

## 4. Conclusions and Outlook

The primary reasons for poor silymarin bioavailability are elevated metabolism, inefficient intestinal absorption, low aqueous solubility, and rapid excretion in bile and urine. These factors necessitate the incorporation of silymarin into a form that can augment its bioavailability. Therefore, silymarin solubilization is fundamental to obtain an optimal and efficient bioavailability and consequently the formulation strategies designed to improve its solubility are of crucial importance. The reviewed studies reveal that increasingly advanced and performing silymarin-based formulations has logically followed step by step the evolution of the nanotechnologies and nanosystems designed to improve the delivery of poorly water-insoluble drugs and active principle ingredients. Current solubilization approaches highlight, for example, the importance and effectiveness of amorphous solid dispersions and lipid-based drug delivery systems, as well as the use of supersaturated solutions. However, most of the formulation preparations suffer some weaknesses such as very complex processes and lack of reproducibility, which are difficult to translate to industrial production. Besides, the drug loading capacity usually is very low and the preparations are unstable during storage. There is still a long way before they come to market. In future, some advanced type of formulation approaches such as emulsomes, mesoporous silica nanoparticles, dendrimers and carbon nanotubes could be successfully utilized for bioavailability enhancement and targeting of silymarin to hepatocytes.

In Table 1 the types of formulations reviewed in the previous sections have been systematically classified and linked to their respective methods of preparation, selected characteristic and related bibliographic sources. We hope that this study could represent a useful reference for a broad and updated overview on the most efficient and relevant nanotechnologies aimed ultimately at improving the therapeutic efficiency of silymarin.

## Figures and Tables

**Figure 1 molecules-24-02155-f001:**
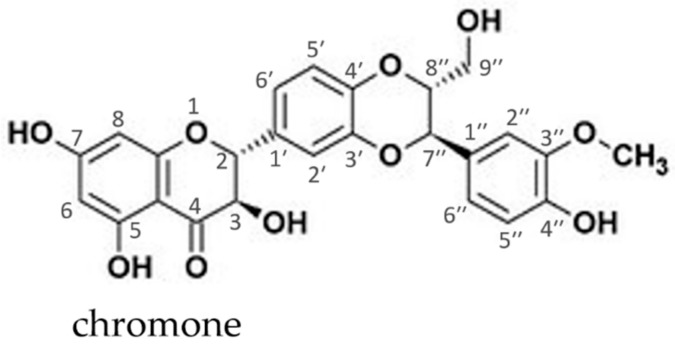
Molecular structure of silybin A.

**Figure 2 molecules-24-02155-f002:**
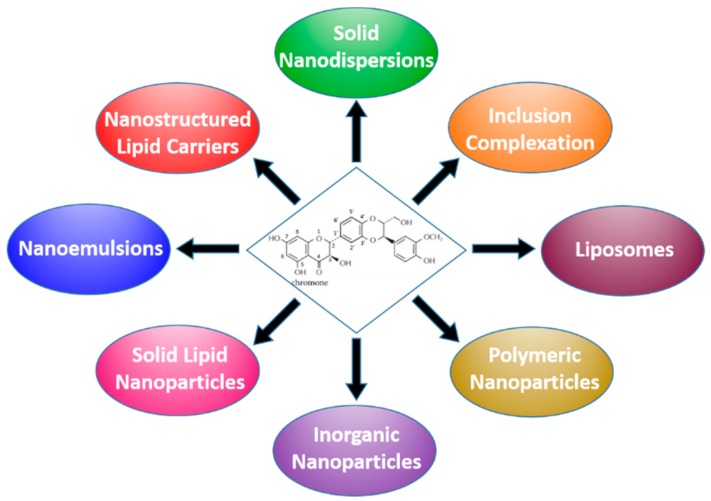
Relevant formulation strategies currently available to improve the bioavailability of silymarin.

**Table 1 molecules-24-02155-t001:** Formulation approaches designed to improve solubility and bioavailability of silymarin.

Type of Formulation	Method of Preparation	Results	References
*Nanocrystals, nanosuspensions*	ESD	Rod-shaped NPs	Zhang et al. [42]
	ESD	NPs < 200 nm	Liu et al. [43]
	HPH	NPs 637 and 132 nm	Whang et al. [44,46,53]
	HPH	Pickering emulsion	Yi et al. [54]
	Spray-drying	Dissolution studies	Hwang et al. [47]
	Spray-drying	Microparticles	Sansone et al. [50]
	Fluid-bed coating	Synchronized release	Wu et al. [48,49]
	Microfluidics	NP size 26–101 nm	Cui et al. [51]
	Antisolvent precip.	Dissolution studies	Sahibzada et al. [52]
	SEDS	Dissolution studies	Yang et al. [55]
*Inclusion complexes, phytosomes*	Co-precipitation	Complex with β-CD	Ghosh et al. [57]
	Freeze-drying	Complex with HP-CD	Kellici et al. [58]
	Kneading	HP-β-CD, RAMEB	Gharbia et al. [59]
	Solvent evaporation	Phospholipids	Yanyu et al. [61]
	Solvent evaporation	Phospholipids	Duan et al. [65]
	Mixed micelles	BS-phospholipids	Yu et al. [63]
	Mixed micelles	BS-phospholipids	Zhu et al. [64]
*Micro- and NanoEmulsions*	Spontaneous emulsif.	Microemulsion	Panapisal et al. [66]
	Low energy emulsif.	O/W emulsion	Abrol et al. [67]
	Low energy emulsif.	O/W emulsion	Parveen et al. [30]
	Low energy emulsif.	Nanoemulsion	Adhikari et al. [74]
	Low energy emulsif.	Nanoemulsion	Calligaris et al. [76]
	Low energy emulsif.	Nanoemulsion	Piazzini et al. [79]
	Membrane emulsif.	Nanoemulsion	Yang et al. [75]
	HPH	Nanoemulsion	Nagi et al. [77]
	SEDDS	Water titration	Wu et al. [70]
	SEDDS	Water titration	Woo et al. [71]
	SEDDS	Water titration	Li et al. [72]
	S-SEDDS	Supersaturated state	Wei et al. [73]
*Liposomes*	Ethanol injection	Drug EE 95%	Maheshwari et al. [82]
	RPE	Drug EE 69%	El-Samaligy et al. [83,84]
	TFD	Drug EE 55%	Kumar et al. [91]
	RPE	Phytosome	Angelico et al. [92,93]
	SEDS	Bile salt	Yang et al. [94]
	TFD	Bile salt	Mohsen et al. [95]
PEGylated liposomes	TFD	Hepatic targeting	Elmowafy et al. [89]
PEGylated liposomes	TFD	Hepatic targeting	Ochi et al. [90]
Proliposomes	Film-deposition	Drug EE 93%	Xiao et al. [85]
Proliposomes	TFD-freeze drying	Drug EE 82%	Tong et al. [87,88]
Cubosomes	Melting/Congealing	Pluronic	Lian et al. [96]
Organogels	Mixed Solution	Lecithin/pluronic	Mady et al. [97]
*Solid-Lipid Nanoparticles*	TFD	Drug EE 99%	Xu et al. [102]
	HPH	Lipolysis mechanism	Shangguan et al. [103]
	Cold/hot HPH	Drug EE 87%	He et al. [104]
	Hot HPH	NP size 165–200 nm	Cengiz et al. [105]
	Hot HPH	SIL-conjugates	Ma et al. [108,109]
	EES	Stealth SLNs	Zhang et al. [106]
	EES	Drug EE 92%	Piazzini et al. [107]
	Film hydration	SIL-emulsomes	Zhou et al. [110]
*Nanostructured Lipid Carriers*	ESE	NP size 230 nm	Jia et al. [111,112]
	ESE	NP size 126 nm	Iqbal et al. [115]
	ESE	NP size 225 nm	Chen et al. [116]
	Hot HPH	Drug EE 87%	Wu et al. [113]
	Emulsif./ultrasound	Drug EE 79%	Chaudhary et al. [114]
Inclusion in polymeric matrices	Co-precipitation	Dissolution studies	Sonali et al. [118]
	Complexation	Chitosan NPs	Nguyen et al. [119]
	ESE/freeze-drying	NP size 100 nm	Zhao et al. [120]
	Nanoprecipitation	Drug EE 79%	Gohulkumar et al. [121]
	Nanoprecipitation	Drug EE 89%	Younis et al. [122]
	Nanoprecipitation	Drug EE 83%	El-Nahas et al. [123]
	Solvent evaporation	Dissolution studies	Yousaf et al. [124]
*Dendrimers and polymeric NPs*	PAMAM dendrimers	Solubility studies	Huang et al. [126]
	PEG-PAMAM	Solubility studies	Diaz et al. [127]
	Polymeric micelles	Chitosan derivative	Sui et al. [128]
	ESE	PLGA	El-Sherbiny [129]
	ESE	PLGA	Snima et al. [130]
	ESE	PLGA	Xie et al. [131]
	ESE	PCL	Bonepally et al. [132]
	Ionic gelation	Chitosan-TPP	Pooja et al. [133]
	Nanoprecipitation	PE Sebacate NPs	Guhagarkar et al. [134]
	Ultrasonication	Polysaccharide NPs	Ma et al. [135]
	Ionic gelation	Chitosan/PGA	Lee et al. [136]
	Ionic gelation	Inulin NPs	Abdel-Wahhab et al. [137]
*Inorganic nanomaterials*	*Microemulsion*	*Mesoporous Si* NP*s*	Cao et al. [139]
	Ultrasonic corrosion	Porous Si NPs	Cao et al. [140,141]
	Drug conjugation	Carbon NT	Tan et al. [143]
	Precipitation	Calcium phosphate	Chen et al. [144]
	Precipitation	Calcium phosphate	Zhu et al. [145]
	Precipitation	PLGA-PEG-Fe_3_O_4_	Ebrahimnezhad et al. [146]
	Coprecipitation	Chitosan-Fe_3_O_4_	Khalkhali et al. [148]
	Emulsion-diffusion	PEG-PLGA-Au	Fazio et al. [149]

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
