# Peer review of "Formulation Strategies for Enhancing the Bioavailability of Silymarin: The State of the Art"

_molecules, 2019, doi:10.3390/molecules24112155_

Round 1

Reviewer 1 Report

The work has a high quality and is of interest to a wide audience.

Author Response

Comments and Suggestions for Authors: The work has a high quality and is of interest to a wide audience

Response: We are very grateful for this comment and appreciate the reviewer has positively evaluated the content and organization of the review.

Reviewer 2 Report

The authors recited a wealth of, but not all, findings related to the topic of silymarin bioavailability. I should say that they have summarized most of the results of interest but overlooked some important facts. The reciting work looks good but there is a lack of in-depth analysis. Moreover, the text reads quite boring.

Silymarin is among the top model drugs I have ever used during the past studies. I choose it because of two reasons--poor solubility as well as poor permeability and the presence of multiple components. It should be born in mind that the pharmacological activity of silymarin is based on the synergistic efficacy of the collection of all components, rather than the main component silybin alone. The fact is that when evaluating the oral bioavailability of silymarin, silybin is generally chosen as an indicative component because it is almost impossible to measure the components other than silybin in silymarin during pharmacokinetic studies. Nevertheless, you need to clarify which component has been used for the evaluation of silymarin bioavailability. The indiscriminative use of "SIL" to represent either silymarin or silybin is misleading. On the other hand, it is incorrect to classify silymarin as a BCS II drugs (poorly soluble but highly permeable). If it is the case, what we need to do is just to solubilize silymarin because the solubilization rather than the permeation across enteric epithelia is the rate-limiting step. The fact that lipid-based vehicles are very efficient in the enhancement of bioavailability speaks that in addition to solubilization, measurements should be taken to enhance the permeability. The balance shifts towards a BCS IV classification for silybin (or silymarin).

It is misleading to just label "nanotechnology" as a measure to enhance the bioavailability of silymarin. We must look beyond the obtained data of bioavailability; we need to dig into the underlying mechanisms. For instance, the lipid-based systems tend to undergo profound lipolysis following ingestion. It is obliged to figure out what happens to the original systems and which secondary or tertiary vehicles function to enhance the permeability. For comparison, a recapitulation of solubilization measures should be introdced first; both solid dispersion and cyclodextrin-related content should be moved to this section. If you dig into the literature, you will find that some researchers did compare various nanovehicles with fast-release formulations. By comparison, the mechanisms beyond solubilization will be highlighted.

It is inappropriate to put the content of mixed micelles or micelles under the subtitle of lipid emulsions. Please exclude the content not related to oral bioavailability, e.g., dermal delivery systems. It is also inappropriate to put both cubosomes and organogels under the subtitle of polymeric matrices; put them under "vesicular vehicles" together with liposomes, or whatever it should be.

The citations are significantly inadequate. Please dig the literature to include as many references as there are. I am sure there are some references "at large".

Please provide a comprehensive table, instead of Table 1, to list the findings.

A few figures can be complemented to make reading more pleasing.

The quote of the composition of silymarin is incorrect. Please refer to works by Ding et al (J Pharm Biomed Anal. 2001;26(1):155-61) and Wu et al (Eur J Pharm Biopharm. 2007;66(2):210-9).

Author Response

Comments and Suggestions for Authors:

Point 1: The authors recited a wealth of, but not all, findings related to the topic of silymarin bioavailability. I should say that they have summarized most of the results of interest but overlooked some important facts. The reciting work looks good but there is a lack of in-depth analysis. Moreover, the text reads quite boring. 

Response 1: We have tried to reorganize the whole paper tacking into account the fruitful suggestions provided by this reviewer, including a new figure summarizing the relevant formulation strategies adopted to enhance the solubility and bioavailability of silymarin. Moreover, the title has been partially modified and more references have been found and added in the revised version of the manuscript, (see the answers to the following points).

Point 2: Silymarin is among the top model drugs I have ever used during the past studies. I choose it because of two reasons - poor solubility as well as poor permeability and the presence of multiple components. It should be born in mind that the pharmacological activity of silymarin is based on the synergistic efficacy of the collection of all components, rather than the main component silybin alone.

The fact is that when evaluating the oral bioavailability of silymarin, silybin is generally chosen as an indicative component because it is almost impossible to measure the components other than silybin in silymarin during pharmacokinetic studies. Nevertheless, you need to clarify which component has been used for the evaluation of silymarin bioavailability. The indiscriminative use of "SIL" to represent either silymarin or silybin is misleading.

Response 2: It is deeply true that the pharmacological activity of an herbal medicine depends on the synergistic activity of a collection of active components in relatively fixed ratios. In the revised version of the manuscript, we mention and illustrate the pioneer works published by prof. W. Wu et al. about the concept of synchronized release in designing sustained or controlled release delivery systems of herbal medicines such as silymarin. Moreover, throughout the text the use of silymarin and silybin has been distinctly recalled for each cited work using the SLM and SIL acronyms, respectively.

Point 3: On the other hand, it is incorrect to classify silymarin as a BCS II drugs (poorly soluble but highly permeable). If it is the case, what we need to do is just to solubilize silymarin because the solubilization rather than the permeation across enteric epithelia is the rate-limiting step. The fact that lipid-based vehicles are very efficient in the enhancement of bioavailability speaks that in addition to solubilization, measurements should be taken to enhance the permeability. The balance shifts towards a BCS IV classification for silybin (or silymarin).

Response 3: The authors acknowledge and comply with the reviewer’s observation. Accordingly, we have corrected the classification from class II to class IV in the new text at lines 130-132.

Point 4: It is misleading to just label "nanotechnology" as a measure to enhance the bioavailability of silymarin. We must look beyond the obtained data of bioavailability; we need to dig into the underlying mechanisms. For instance, the lipid-based systems tend to undergo profound lipolysis following ingestion. It is obliged to figure out what happens to the original systems and which secondary or tertiary vehicles function to enhance the permeability.

Response 4: We acknowledge the reviewer for his relevant observation. We reply by remarking that this review article has been written with the following aims: To describe and illustrate the evolution of formulation strategies for enhancing the solubility and bioavailability of silymarin, with a focus on latest nanotechnological approaches. While complying the correct observation of looking beyond the obtained data about bioavailability, and re-discussing in terms of mechanism of action of various formulations, it is remarked that this is outside the main purpose of the present revision whose objective is instead to provide a vision as comprehensive as possible on modern formulation technologies designed for silymarin.

Point 5: For comparison, a recapitulation of solubilization measures should be introduced first; both solid dispersion and cyclodextrin-related content should be moved to this section. If you dig into the literature, you will find that some researchers did compare various nanovehicles with fast-release formulations. By comparison, the mechanisms beyond solubilization will be highlighted.

Response 5: We acknowledge the reviewer and accordingly the order of illustration of various types of formulations has been changed in the revised version, and new cited references have been added in the right places.

Point 6: It is inappropriate to put the content of mixed micelles or micelles under the subtitle of lipid emulsions. Please exclude the content not related to oral bioavailability, e.g., dermal delivery systems.

It is also inappropriate to put both cubosomes and organogels under the subtitle of polymeric matrices; put them under "vesicular vehicles" together with liposomes, or whatever it should be.

Response 6: Although most of the reported investigations consider studies designed to enhance silymarin bioavailability via oral administration, our goal is to provide a broad overview of all the applications concerning the formulations based on silymarin, with no exception regarding, e.g., injectable and transdermal delivery systems. Moreover, accepting the reviewer’s suggestion, the cited works about the use of mixed micelles have been moved to paragraph 3.2 of the revised version, whereas cubosomes and organogels now appear at the end of the subparagraph 3.3.2.

Point 7: The citations are significantly inadequate. Please dig the literature to include as many references as there are. I am sure there are some references "at large".

Response 7: The number of references has been increased from 127 to 150 in the revised version of the paper.

Point 8: Please provide a comprehensive table, instead of Table 1, to list the findings.

Response 8: Table 1 has been completely redone and updated taking into account the changes made in the text and the inclusion of new bibliographical references.

Point 9: A few figures can be complemented to make reading more pleasing.

Response 9: To comply with the reviewer’s observation a new figure 2 has been added to paragraph 3 illustrating as a cartoon the relevant formulation strategies currently adopted to improve the bioavailability of silymarin.

Point 10: The quote of the composition of silymarin is incorrect. Please refer to works by Ding et al. (J Pharm Biomed Anal. 2001;26(1):155-61) and Wu et al. (Eur J Pharm Biopharm. 2007;66(2):210-9).

Response 10: The recommended references have been added as [17,18] and accordingly the silymarin composition has been modified in the text (line 105).

Reviewer 3 Report

In this work, Costanzo et al. reviewed the progress in nanoparticle strategies for enhancing the bioavailability of silymarin. The manuscript is fairly interesting but needs extensive revision before publication.

The language of this manuscript needs to be extensively improved with the help of professional editing.

It is recommended in each section, a graph showing the representative important research work should be added.

In the Introduction section, a general brief summary of the types of nanocarriers that has been developed and how they have been modified to achieve better in vitro and in vivo fate should be discussed. Recommended literature: Advanced Drug Delivery Reviews 2019 (https://doi.org/10.1016/j.addr.2019.01.002); Clinical Translational Medicine 2017 (https://doi.org/10.1186/s40169-017-0175-0) et al.

The Conclusion and outlook section is too brief. The authors should provide more insights on the current and future of the research area.

Author Response

Comments and Suggestions for Authors:

Point 1: In this work, Costanzo et al. reviewed the progress in nanoparticle strategies for enhancing the bioavailability of silymarin. The manuscript is fairly interesting but needs extensive revision before publication. The language of this manuscript needs to be extensively improved with the help of professional editing.

Response 1: We are grateful to the reviewer for his positive comment about our work. Syntax and prose have been extensively reviewed by a native speaking English, which hopefully will render more readable the whole paper.

Point 2: It is recommended in each section, a graph showing the representative important research work should be added.

Response 2: In accordance with the reviewer's suggestion, in the revised version of the manuscript each paragraph now contains a brief description of the specific formulation that will be discussed, supported also by qualified cited reviews on the subject.

Point 3: In the Introduction section, a general brief summary of the types of nanocarriers that has been developed and how they have been modified to achieve better in vitro and in vivo fate should be discussed. Recommended literature: Advanced Drug Delivery Reviews 2019 (https://doi.org/10.1016/j.addr.2019.01.002); Clinical Translational Medicine 2017 - (https://doi.org/10.1186/s40169-017-0175-0) et al.

Response 3: The Introduction has been completely rewritten according to the reviewer’s suggestion. Moreover, the authors acknowledge the reviewer for his comments since these give us the possibility to improve and update the literature overview. Thus, the recommended references have been added as [14] and [15] in the new version of the paper.

Point 4: The Conclusion and outlook section is too brief. The authors should provide more insights on the current and future of the research area.

Response 4: The final Conclusion section has been rewritten trying to underline several aspects of the future trends in the formulation technology of poor water soluble plant extract such as silymarin.

Round 2

Reviewer 2 Report

The quality of this manuscript has been improved significantly after revision. It is acceptalbe in its current form.

Reviewer 3 Report

The authors have addressed my concerns.